# On Space Folds of ReLU Neural Networks

**Michal Lewandowski**                                               *michal.lewandowski@scch.at*
*Software Competence Center Hagenberg (SCCH)*

**Hamid Eghbalzadeh**                                                *heghbalz@meta.com*
*AI at Meta*

**Bernhard Heinzl**                                                  *bernhard.heinzl@scch.at*
*Software Competence Center Hagenberg (SCCH)*

**Raphael Pisoni**                                                   *raphael.pisoni@scch.at*
*Software Competence Center Hagenberg (SCCH)*

**Bernhard A.Moser**                                                 *bernhard.moser@scch.at*
*Software Competence Center Hagenberg (SCCH)*
*Johannes Kepler University of Linz (JKU)*

**Reviewed on OpenReview:** *https://openreview.net/forum?id=RfFqBXLDQk*

## Abstract

Recent findings suggest that the consecutive layers of ReLU neural networks can be understood geometrically as space folding transformations of the input space, revealing patterns of self-similarity. In this paper, we present the first quantitative analysis of this space folding phenomenon in ReLU neural networks. Our approach focuses on examining how straight paths in the Euclidean input space are mapped to their counterparts in the Hamming activation space. In this process, the convexity of straight lines is generally lost, giving rise to non-convex folding behavior. To quantify this effect, we introduce a novel measure based on range metrics, similar to those used in the study of random walks, and provide the proof for the equivalence of convexity notions between the input and activation spaces. Furthermore, we provide empirical analysis on a geometrical analysis benchmark (CantorNet) as well as an image classification benchmark (MNIST). Our work advances the understanding of the activation space in ReLU neural networks by leveraging the phenomena of geometric folding, providing valuable insights on how these models process input information.

## 1 Introduction

Neural networks are inspired by the biological structure of the brain (Rosenblatt, 1958). They achieve outstanding performance across various domains, including computer vision (Krizhevsky et al., 2012) and speech recognition (Maas et al., 2013). However, despite these impressive results, their underlying mechanisms remain poorly understood from a mathematical perspective, and current advances lack a solid foundation in rigorous mathematical analysis (Zhang et al., 2017; Neyshabur et al., 2017; Marcus, 2018; Sejnowski, 2020).

As we will discuss in this work, geometric folding can provide valuable insights and serve as a useful tool to expand our understanding of neural networks. The phenomena of geometric folding can be described as the process by which a structure undergoes a transformation from a linear or planar form into a more compact, layered configuration, where space is efficiently organized through recursive bending or folding patterns. For example, in biological systems, DNA molecules fold into complex yet highly organized shapes to fit within the confines of a cell nucleus (Dekker et al., 2013). Further, also proteins fold into precise, three-dimensional shapes, transforming from linear amino acid chains into complex structures essential for their specific functions (Crescenzi et al., 1998; Dill et al., 2008; Jumper et al., 2021). Folding has been argued

to appear in neural networks, as the layering of data representations across the network depth allows for increasingly abstract, compact, and hierarchical information encoding, capturing patterns at multiple scales. It was proposed more than a decade ago, that successive layers of ReLU neural networks can be interpreted as folding operators (Montúfar et al., 2014; Raghu et al., 2017). These folds result in the replication of shapes formed by the network and contribute to understanding how the space is folded, which can help reveal symmetries in the decision boundaries that the network learns. Keup and Helias (2022) likened this process to the physical process of paper folding, where the input space is "folded" during learning. However, folding occurred in neural networks is elusive in the continuous input space. In case of protein folding, this process has been quantified using discrete mathematics on sequences of amino acids (Crescenzi et al., 1998). For neural ReLU networks, the activation space offers a possibility for further analyses of this phenomenon. In this paper, we focus on the activation space to investigate symmetries and self-similarity in the learned regions (Balestriero et al., 2024). The activation space and its related linear regions have also been used also as a measure of the network's expressivity (e.g., Montúfar et al. (2014); Raghu et al. (2017); Hanin and Rolnick (2019a)).

Insofar, the concept of space folding by neural networks remains largely qualitative, with no prior work attempting to quantify these effects. In this paper, we introduce the first method for measuring these transformations using range measures (Weyl, 1916; Moser, 2012) and discrete mathematics. Our analysis is based on a topological investigation of the activation patterns along a straight path in the activation space. In the Euclidean space the shortest path between two points is a straight line. Walking along such a path without turns monotonically increases the distance to the starting point. However, this observation no longer applies in the activation space. During the folding operation, the convexity of the created linear regions (defined in Sec. 3), may not be preserved and the Hamming distance on a straight path between two (non-adjacent) patterns can decrease. This lack of preservation inspires the introduction of our space folding measure, which measures the deviations from convexity on a straight path in both the input (Euclidean) and the activation (Hamming) spaces. In summary, our contributions in this work are as follows:

- We prove the equivalence of convexity notions between the input and activation spaces.

- We introduce a *space folding* measure to quantify local deviations from convexity in the activation space of ReLU networks. We provide both local and global versions of our measure. We further provide the exact algorithm for its computation, together with its complexity analysis, and suggest the ways of reducing this complexity through intra-class clustering of samples.

- We experimentally investigate the behaviour of our measure on (*i*) CantorNet, a specially constructed synthetic example with an arbitrarily ragged decision surface, and (*ii*) ReLU networks with varying depth and width with constant number of hidden neurons trained on the MNIST benchmark. We then extend the analysis to larger networks and find that, although the folding values do not change much, the ratio of inter-digit paths that exhibit the folding effects increases significantly.

The remainder of the paper is organized as follows: Sec. 2 details the related work; Sec. 3 recalls some basic facts and fixes notation for the rest of the paper; Sec. 4 establishes convexity results, Sec. 5 introduces the space folding measure; Sec. 6 describes the experimental results and discusses how the measure extends to different types of layers; Sec. 7 outlines the future work; Sec. 8 provides concluding remarks, the take-away message of our work and possible future directions for our work. In Appendices, Appendix A briefly introduces CantorNet (following Lewandowski et al. (2024)); Appendix B details results obtained on the MNIST dataset; Appendix C provides a preliminary study of the sensitivity of measure to intra-class clustering of samples; Appendix D presents the results obtained for larger networks.

## 2 Related work

**Activation Space.** The pioneering study by Makhoul et al. has investigated partitioning the input space with neural networks with 2-hidden-layers, thresholding neurons with the ReLU activation function (without naming it). With two hidden layers, the first layer creates hyperplanes that divide the input space into regions, adjacent if they have a Hamming distance of 1 (Makhoul et al., 1991). Connected regions are

defined by a path through adjacent regions. The interest in the number of these regions was revived in 2014 by Montúfar et al., with several follow-up works, e.g., (Raghu et al., 2017; Serra et al., 2018; Xiong et al., 2020; Hanin and Rolnick, 2019a;b). The authors provided ever tighter bounds on the number of activation regions, and used them as a proxy for its expressiveness, among others.

**Space Folds.**   The idea of folding the space has been investigated in the computational geometry (Demaine et al., 2000). Demaine and Demaine surveyed the phenomenon, focusing on the type of object being folded, e.g., paper, or polyhedra. Bern and Hayes explored whether a given crease pattern can be folded into a flat origami (non-crossing polygons in 2D with layers). Later, Bern and Hayes showed that any compact, orientable, piecewise-linear 2-manifold with a Euclidean metric can achieve this structure. In Montúfar et al. (2014) in Section 2.4, the authors briefly mentioned the folding phenomena, although through the lens of linear regions. They argue that each hidden layer in a neural network acts as a folding operator, recursively collapsing input-space regions. This folding depends on the network's weights, biases, and activation functions, resulting in input regions that vary in size and orientation, highlighting the network's flexible partitioning. In Phuong and Lampert (2020), in the Appendix A.2 the authors explored the folding operation by ReLU neural networks, but leave the exploration quite early on. In Keup and Helias (2022), the authors argued that it is through the folding operation that the neural networks arrive at their approximation power.

**Self-Similarity and Symmetry.**   Self-similarity and symmetry are related but distinct concepts, often found in nature, mathematics, and physics. Self-similarity means that a structure or pattern looks similar to itself at different scales, and is also present in numerical data, e.g., images (Wang et al., 2020), audio tracks (Foote, 1999) or videos (Alemán-Flores and Álvarez León, 2004). Symmetry implies that an object or pattern is invariant under certain transformations, e.g., reflection, rotation, or translation. In the context of neural networks, in Grigsby et al. (2023), the authors describe a number of mechanisms through which hidden symmetries can arise. Their experiments indicate that the probability that a network has no hidden symmetries decreases towards 0 as depth increases, while increasing towards 1 as width and input dimension increase. Many fractal shapes, such as the Mandelbrot set (Mandelbrot, 1983) or CantorNet (Lewandowski et al., 2024), exhibit both self-similarity and certain symmetries. Moreover, both concepts relate to the folding operation: invariance under reflection (symmetry) can be equivalently understood as a space fold, and self-similarity can be interpreted as recursive folding or scaling operations that replicate the pattern across different levels. In neural network architectures, these principles can manifest through hierarchical structures, where each layer effectively "folds" information from previous layers, producing patterns that may repeat or reflect across layers or nodes (Raghu et al., 2017).

**Distance Alteration.**   Lipschitz constant, a well established concept in mathematical analysis, bounds how much function's output can change in proportion to a change in its input. In context of neural networks, it has been linked to adversarial robustness, e.g., (Tsuzuku et al., 2018; Virmaux and Scaman, 2018), or generalization properties, e.g., (Bonicelli et al., 2022). Cisse et al. showed that the Lipschitz constant of a neural network can grow exponentially with its depth. Anil et al. observe that enforcing the Lipschitz property leads to some limitations, and show that norm-constrained ReLU networks are less expressive than unconstrained ones. The exact computation of the Lipschitz constant, even for shallow neural networks (two layers), is NP-hard (Virmaux and Scaman, 2018). In Gamba et al. (2023), the authors experimentally studied the (empirical) Lipschitz constant of deep networks undergoing double descent, and highlighted non-monotonic trends strongly correlating with the test error. Finally, Hanin et al. prove that the expected length distortion slightly shrinks for ReLU networks with standard random initialization, building on the results of Price and Tanner. While important, none of the aforementioned works touch on the activation space of neural networks, nor do they investigate the monotonicity of a mapped straight line. Our analysis extends beyond the concept of the Lipschitz constant by exploring the convolution of the input space under a neural network.

## 3   Preliminaries

We define a *ReLU neural network* $\mathcal{N} : \mathcal{X} \rightarrow \mathcal{Y}$ with the total number of $N$ neurons as an alternating composition of the ReLU function $\sigma(x) := \max(x, 0)$ applied element-wise on the input $x$, and affine functions

Euclidean Input Space 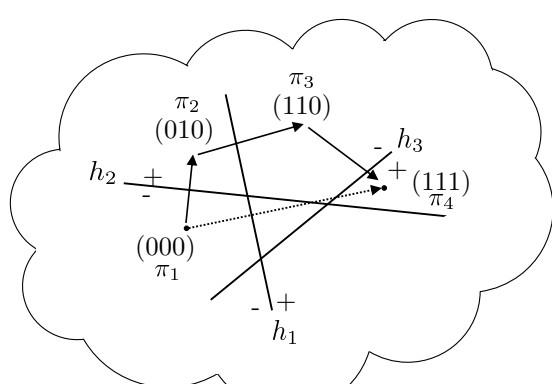   Hamming Activation Space 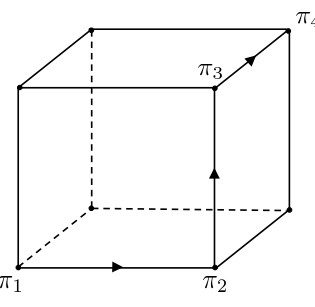

Figure 1: Illustration of a walk on a straight path in the Euclidean input space and the Hamming activation space. Left: the dotted line represent the shortest path in the Euclidean space. The arrows represent *a* shortest path in the Hamming distance between activation patterns $\pi_1$ and $\pi_4$ (note that in the Hamming space the notion of the shortest path becomes ambiguous). Right: The illustration of a shortest path connecting $\pi_1$ and $\pi_4$ in the Hamming activation space.

with weights $W_k$ and biases $b_k$ at layer $k$. An input $x \in \mathcal{X}$ propagated through $\mathcal{N}$ generates non-negative activation values on each neuron. A *binarization* is a mapping $\pi : \mathbb{R}^N \to \{0,1\}^N$ applied to a vector $v = (v_1, \dots, v_N) \in \mathbb{R}^N$, resulting in a binary vector by clipping strictly positive entries of $v$ to 1, and non-positive entries to 0, that is $\pi(v_i) = 1$ if $v_i > 0$, and $\pi(v_i) = 0$ otherwise. In our case, the vector $v$ is the concatenation of all neurons of all hidden layers, called an *activation pattern*, and it represents an element in a binary hypercube $\mathcal{H}_N := \{0,1\}^N$ where the dimensionality is equal to the number $N$ of hidden neurons in network $\mathcal{N}$. A *linear region* is an element of a partition covering the input domain where the network behaves as an affine function (Montúfar et al., 2014) (see Fig. 1, left). The Hamming distance, $d_H(u, v) := |\{u_i \neq v_i \text{ for } i = 1, \dots, N\}|$, measures the difference between $u, v \in \mathcal{H}_N$.

## 4 Convexity

Convexity is a key concept in computational geometry and plays a critical role in various computer engineering applications, such as robotics, computer graphics, and optimization (Boissonnat and Yvinec, 1998). In Euclidean space, convexity can be defined as a property of sets that are closed under convex combinations, where the set contains all line segments between any two points within it (Roy and Stell, 2003). We extend this notion of convexity to the Hamming space as follows.

**Definition 1** (Adapted from Moser et al. (2022)). *A subset $S$ of the Hamming cube $\mathcal{H}^n$ is convex if, for every pair of points $x, y \in S$, all (observable) points on every shortest path between $x, y$ are also in $S$.*[1]

**Example 1.** *Consider points $\pi_1 = (000)$ and $\pi_4 = (111)$ (Fig. 1, right). Then the Hamming distance $d_H(\pi_1, \pi_4) = 3$. Every shortest path consists of three edges, flipping one "bit" at a time, and thus a convex set is the whole Hamming cube $\mathcal{H}^3$.*

**Example 2.** *Consider activation patterns $\pi_1 = (0111), \pi_2 = (0001), \pi_3 = (1011)$ as shown in Fig. 2 (see Appendix A for more details). For a walk through any two of the activation patterns, (1) $\pi_1 \to \pi_2$, (2) $\pi_2 \to \pi_3$, (3) $\pi_1 \to \pi_3$, there are intermediate activation patterns $\pi_{inter}$ that we traverse. They are, respectively: (1) $\pi_{inter} = \{(0011), (0101)\}$, (2) $\pi_{inter} = \{(0011), (1001)\}$, (3) $\pi_{inter} = \{(0011), (1111)\}$. As none of them is contained in $\{\pi_4, \pi_5, \pi_6\}$, we conclude that they are non-observable on the considered domain, $[0,1] \times [0,1]$. Thus, the activation patterns $\{\pi_1, \pi_2, \pi_3\}$ form a convex set in the Hamming cube sense.*

---

[1]In the discrete activation space it might happen that some points (i.e., binary vectors) are non-observable (e.g., point (011) for the tessellation in Fig. 1).

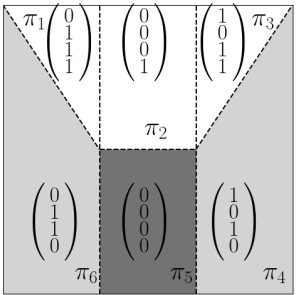

Figure 2: Activation patterns $\pi_i$ of recursion-based representation of CantorNet (see Appendix A). We skip neurons with unchanged values. The colours are used for increased visibility; activation patterns $\{\pi_1, \pi_2, \pi_3\}$ are convex in the Hamming cube sense (see Ex. 2). The darker gray of $\pi_5$ has been used to visually distinguish from $\pi_4$ and $\pi_6$. (Adapted from Lewandowski et al. (2024) with the authors' approval.)

Before introducing the space folding measure, which relies on the notion of convexity, we prove the equivalance of convexity notions between the input and activation spaces for hyperplanes that intersect the entire input space. This further justifies the need of deeper layers to observe any space folding effects.

**Lemma 1.** *Consider a tessellation of activation regions formed by $N$ hyperplanes $h_1, \ldots, h_N$ with activation regions $R_{\pi_1}, \ldots, R_{\pi_r} \subset \mathbb{R}^n$ and corresponding activation patterns $\mathcal{A} = \{\pi_1, \ldots, \pi_r\}$. A union $R = \bigcup_{\pi \in \mathcal{A}} R_\pi$ of activation regions is convex in $\mathbb{R}^n$ if and only if the set $\mathcal{A}$ of corresponding activation patterns is convex in the Hamming space $\mathcal{H}^m$.*

*Proof.* Convexity in $\mathbb{R}^n \Rightarrow$ Convexity in Hamming space:

Assume that the set $R = \bigcup_{\pi \in \mathcal{A}} R_\pi$ is convex in $\mathbb{R}^n$. We want to show that the set $\mathcal{A}$ is convex in the Hamming space. We start with showing the connectivity of $\mathcal{A}$. Let $\pi_i, \pi_j \in \mathcal{A}$ be any two activation regions, and choose any points $P \in R_{\pi_i}$ and $Q \in R_{\pi_j}$ in respective activation regions. Since $R$ is convex, the line segment $[P, Q]$ lies entirely within $R$. As we move along $[P, Q]$, we may cross hyperplanes $h_k$ where the activation state changes. Each such crossing corresponds to flipping exactly one bit in the activation pattern (see Fig. 3). This sequence of bit flips forms a path in the Hamming space from $\pi_i$ to $\pi_j$, showing that $\mathcal{A}$ is connected. Let us now show the convexity of $\mathcal{A}$. Assume, for contradiction, that $\mathcal{A}$ is not convex in the Hamming space. Then, there exists a shortest path $\gamma$ in the Hamming space connecting $\pi_i$ and $\pi_j$ that leaves $\mathcal{A}$; that is, some activation patterns along $\gamma$ are not in $\mathcal{A}$. However, from connectivity, the path corresponding to the line segment $[P, Q]$ stays entirely within $\mathcal{A}$, as it corresponds to activation patterns of points within $R$. Since $[P, Q]$ is a straight line, it corresponds to a minimal sequence of bit flips (i.e., a shortest path in the Hamming space). Therefore, there exists a shortest path within $\mathcal{A}$, contradicting the assumption. Hence, $\mathcal{A}$ is convex in the Hamming space.

Convexity in Hamming space $\Rightarrow$ Convexity in $\mathbb{R}^n$:

Now assume that the set $\mathcal{A}$ of activation patterns is convex in the Hamming space. We want to show that the union $R = \bigcup_{\pi \in \mathcal{A}} R_\pi$ is convex in $\mathbb{R}^n$. Let $P, Q \in R$ be any two points, and denote by $R_{\pi_1}, R_{\pi_3} \in \mathcal{A}$ their activation patterns. Consider the line segment $[P, Q]$ in $\mathbb{R}^n$. As we move from $P$ to $Q$, we may cross hyperplanes $h_k$, changing activation patterns. Each crossing of a hyperplane $h_k$ corresponds to flipping a bit in the activation pattern, forming a path in the Hamming space from $\pi_P$ to $\pi_Q$, as previously (again, see Fig. 3). Since $\mathcal{A}$ is convex in the Hamming space, all shortest paths between $\pi_P$ and $\pi_Q$ remain within $\mathcal{A}$, and in particular, the sequence of activation patterns along $[P, Q]$ is such a shortest path. Therefore, all activation patterns along $[P, Q]$ are in $\mathcal{A}$. Since every point along $[P, Q]$ has an activation pattern in $\mathcal{A}$, it lies within $R$. Thus, $[P, Q] \subset R$, showing that $R$ is convex in $\mathbb{R}^n$. $\qquad\square$

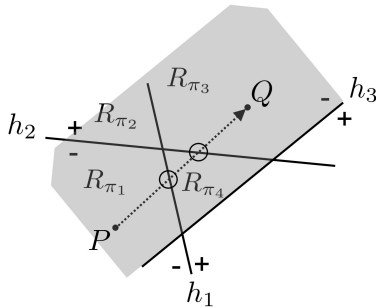

Figure 3: The shaded gray area illustrates a convex set in the Euclidean space. The hyperplanes $h_1, h_2, h_3$ intersect the entire input space (it holds for the hyperplanes described by neurons from the first hidden layer of a ReLU neural network). A straight line $[P, Q]$ connecting points $P$ and $Q$ crosses hyperplanes $h_1$ and $h_3$, resulting in a "bit" flip at a time.

Lemma 1 is important for two reasons. First, its direct consequence is that the space folding effects are observable only in networks with at least two hidden layers. Moreover, it proposes another angle to see the classic XOR problem (Minsky and Papert, 1969), and why one layer is insufficient for class separation. Second, observe limitations of Lemma 1: We assume that along a walk on a shortest path (in the Hamming sense) between any activation patterns corresponding to linear regions in a convex arrangement $\mathcal{A}$ in the Euclidean input space, the Hamming distance of adjacent linear regions differs by one, which only holds in case of hyperplanes that intersect the entire input space, and such hyperplanes are described by the $1^{\text{st}}$ hidden layer of a ReLU neural network (e.g., Raghu et al. (2017)). For deeper layers, there may appear activation regions, which, although neighboring, may have the Hamming distance exceeding one (see Example 2).

## 5  Analysis in the Activation Space

**Range Measures.**  Our space folding measure is inspired by the construction of range measures of random walks. Consider a walk along a line given by the sequence $s = (s_k)_{k=0}^N$ of $N$ steps of length $s_i$ at $i^{\text{th}}$ step. At each step $i$ the walk can go either up or down. The maximum absolute route amplitude of the walk is given by $r_A(s) := \max_{n \leq N} |\sum_{j=0}^n s_j|$, dependent on the choice of the coordinate system. However, we can remove this dependency by considering $r_D(s) := \max_{n \leq N}\{\sum_{k=0}^n s_k, 0\} - \min_{n \leq N}\{\sum_{k=0}^n s_k, 0\} = \max_{k \leq n \leq N} |\sum_{j=k}^n s_j|$, which represents the diameter of the walk invariant under translational coordinate transformations (Moser, 2014; 2017). Both $r_A(s)$ and $r_D(s)$ are examples of range metrics encountered in the field of random walks in terms of an asymptotic distribution resulting from a diffusion process (Jain and Orey, 1968). It turns out that $r_A(s)$ and $r_D(s)$ are norms, $\|s\|_A$ and $\|s\|_D$, respectively (Alexiewicz, 1948). Note that $\|s\|_A \leq \|s\|_D \leq 2\|s\|_A$, stating the norm-equivalence of $\|.\|_A$ and $\|.\|_D$.

**Example 3.** *A simple example of a range measure is a variance estimator of a sample with n observations,* $\mathbf{x} = (x_1, \ldots, x_n)$, *defined as* $\text{Var}(\mathbf{x}) = \frac{1}{n(n-1)} \sum_{i=1}^n (x_i - \bar{x})^2$, *where* $\bar{x}$ *is the sample's arithmetic average.*

**The Space Folding Measure.**  Consider a straight line connecting two samples $\mathbf{x}_1, \mathbf{x}_2$ in the Euclidean input space realized as a convex combination $(1 - \lambda_i)\mathbf{x}_1 + \lambda_i \mathbf{x}_2$, where $\lambda_i$ are equally spaced on $[0, 1]$ (the equal spacing is due to practicality, but is not necessary). Then, consider the mapping of the straight line $[\mathbf{x}_1, \mathbf{x}_2]$ to a *path* $\Gamma$ in the Hamming activation space, with intermediate points $(\pi_1, \ldots, \pi_n)$, $\pi_i \in \mathcal{H}^N$ under a neural network $\mathcal{N}$ (see Fig. 4, left). We consider a change in the Hamming distance at each step $i$

$$\Delta_i := d_H(\pi_{i+1}, \pi_1) - d_H(\pi_i, \pi_1). \tag{1}$$

We then look at the maximum of the cumulative change $\max_k \sum_{i=1}^{k} |\Delta_i|$ along the path $\Gamma$,

$$r_1(\Gamma) = \max_{i \in \{1,\dots,n-1\}} \sum_{j=1}^{i} (d_H(\pi_{j+1}, \pi_1) - d_H(\pi_j, \pi_1)) = \max_{i \in \{1,\dots,n-1\}} d_H(\pi_i, \pi_1). \tag{2}$$

The above expression equals to the maximum Hamming distance to the starting point reached along the path. Next, we keep track of the total distance traveled on the hypercube when following the path,

$$r_2(\Gamma) = \sum_{i=1}^{n-1} d_H(\pi_i, \pi_{i+1}). \tag{3}$$

For the measure of space flatness, we consider the ratio $\phi(\Gamma) := r_1(\Gamma)/r_2(\Gamma)$. Equivalently, the space folding measure equals

$$\chi(\Gamma) := 1 - \phi(\Gamma) = 1 - \max_{i \in \{1,\dots,n\}} d_H(\pi_i, \pi_1) \Big/ \sum_{i=1}^{n-1} d_H(\pi_i, \pi_{i+1}). \tag{4}$$

Lemma 1 guarantees that a straight line in both the input and the activation space is convex, and $\chi$ measures the deviation from convexity along this path, effectively measuring deviation from flatness, hence its name. The higher $\chi$ is, the more folded the space is along the path $\Gamma$. We say that the space is flat if it is not folded, and in that sense "folding" is opposite to "flatness".

**Lemma 2.** *For every path $\Gamma$ the space folding measure satisfies $0 \leq \chi(\Gamma) \leq 1$ (provided that $\sum_{i=1}^{n-1} d_H(\pi_i, \pi_{i+1}) > 0$, i.e., the path $\Gamma$ traverses more than one region).*

*Proof.* We only show the upper bound as the lower is obtained in the similar way. From the triangle inequality for any activation patterns $\pi_1, \pi_i, \pi_{i+1} \in \mathcal{H}^N$ it holds that $d_H(\pi_i, \pi_{i+1}) \leq d_H(\pi_i, \pi_1) + d_H(\pi_1, \pi_{i+1})$. Writing this for every index $i \in \{1, \dots, n-1\}$, $n > 2$, and summing by sides we obtain

$$\sum_{i=1}^{n-1} d_H(\pi_i, \pi_{i+1}) \leq \sum_{i=1}^{n-1} d_H(\pi_i, \pi_1) + \sum_{i=1}^{n-1} d_H(\pi_1, \pi_{i+1}) \leq 2(n-1) \max_i d_H(\pi_i, \pi_1).$$

Recall that $\sum_i d_H(\pi_i, \pi_{i+1}) > 0$ and divide each side by this sum. It follows that

$$\chi(\Gamma) \leq 1 - \frac{1}{2(n-1)} \leq 1.$$

$\square$

To understand the motivation behind the construction of the measure, consider a straight path $\Gamma$ in the Euclidean input space that gets mapped to a straight path in the Hamming space. In this case, the range measures $r_1, r_2$ increase and their ratio $r_1(\Gamma)/r_2(\Gamma) = 1$, thus there is no space folding, i.e., $\chi(\Gamma) = 0$. If a straight path in the Euclidean path gets mapped to a curved path in the Hamming activation space (Fig. 4, left), we observe non-zero values of the space folding measure $\chi$. The space folding can equal $\chi(\Gamma) = 1$ in the case presented in Fig. 4, right. Consider a path $\Gamma = (\pi_1, \pi_2, \pi_1, \pi_2, \dots)$. Then, the range measure $r_1(\Gamma) = 1$ and $r_2(\Gamma) \to \infty$, hence $\chi(\Gamma) \to 1$. Our measure can be made global by considering the supremum over all possible paths $\Gamma$ in the Hamming activation space, i.e.,

$$\Phi_{\mathcal{N}} := \sup_{\Gamma \in \mathcal{X}} \chi(\Gamma). \tag{5}$$

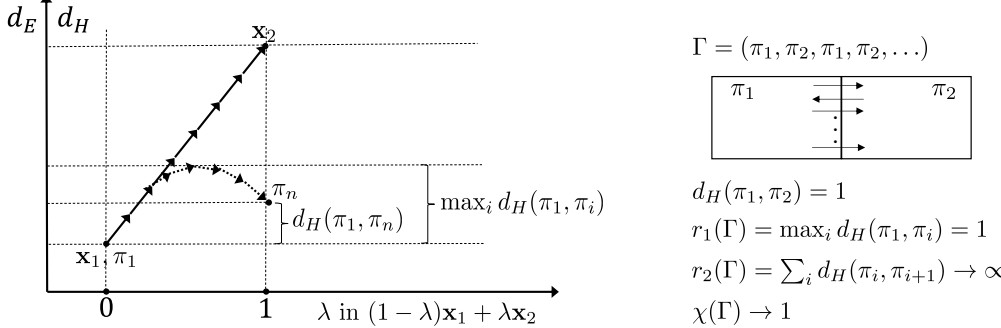

Figure 4: **Left**: Straight line between $\mathbf{x}_1$ and $\mathbf{x}_2$ in the Euclidean space. Observe that, when mapped to the Hamming activation space (dotted arrows), the Hamming distance may decrease when following the path, i.e., it might happen that $d_H(\pi_1, \pi_n) < \max_i d_H(\pi_1, \pi_i)$. **Right**: An extreme case when space folding $\chi(\Gamma) = 1$. Note that it is sufficient that $r_1(\Gamma) = c$ for some $c \in \mathbb{R}_+$, and that the path $\Gamma$ is looped between the same regions, resulting in $r_2(\Gamma) \to \infty$. This construction, although theoretically possible, might not be realizable in practice.

**The Algorithm.** In this paragraph, we present the algorithm for computing the space folding measure $\chi$ introduced earlier, along with its associated computational complexity.

---

**Algorithm 1:** Computation of the Space Folding Measure (Eq. 4)

---

**Input:** Two input samples $\mathbf{x}_1$, $\mathbf{x}_2$, the number of intermediate points $n$, the total number of hidden
       neurons $N$, cost of running the network in the inference mode $O(\texttt{C})$
**Output:** Space Folding $\chi(\Gamma)$ as in Eq. 4
**Step 1:** Linearly interpolate $\mathbf{x}_1$ and $\mathbf{x}_2$, sampling $n$ points;       // Sampling Complexity:  $O(n)$
**Step 2:** For each sampled point:
**begin**
    | Compute the binarisation;       // Binarization Complexity Per Point:  $O(\texttt{C})$
// Total Binarization Complexity:  $O(n \cdot \texttt{C})$
**Step 3:** Compute the maximal (from the starting point) and total Hamming distances between
 intermediate points;       // Computation of Range Measures Complexity:  $O(n \cdot N)$
**return** Space Folding $\chi(\Gamma)$;       // Total Algorithm Complexity:  $O\left(n \cdot (N + \texttt{C})\right)$

---

In the comments for each line, we have included the computational complexity of the corresponding operation. The complexity of running the network in inference mode is denoted as $O(\texttt{C})$. The total computational complexity for every pair of samples from classes $C_1$ and $C_2$ is thus

$$O\left(n \cdot (N + \texttt{C}) \cdot |C_1| \cdot |C_2|\right), \tag{6}$$

where $|C_1|, |C_2|$ denote the cardinalities of the respective classes of points $\mathbf{x}_1, \mathbf{x}_2$. Since $N + \texttt{C} = \text{const}$, the overall computational cost can be controlled by adjusting $n$ or by subsampling within the classes. In the following, we establish an upper bound on the number of intermediate steps $n$. For a neural network with $N$ hidden neurons, consider two input samples $\mathbf{x}_1$ and $\mathbf{x}_2$. The maximum Hamming distance between their corresponding activation patterns, $\pi_1$ and $\pi_2$, equals $d_H(\pi_1, \pi_2) = N$. Consequently, the number of linear regions between inputs $\mathbf{x}_1$ and $\mathbf{x}_2$ is at most $N$. Since intermediate steps that fall within the same activation region do not affect the space folding measure, the number of steps $n$ should not exceed $N$. We note that evenly spacing intermediate points for the computation of the space folding measure $\chi$ may be suboptimal. Instead, the focus should be on traversing *all* linear regions between the two samples. Optimization of the path will be addressed in a future submission.

Secondly, to address the (potentially) high cardinality of classes $C_1$ and $C_2$, we propose the following approach. Let $m_1 < |C_1|$ and $m_2 < |C_2|$, where $m_1$ and $m_2$ represent the number of clusters into which the samples in classes $C_1$ and $C_2$ are grouped. For each cluster in respective class $C_1$ or $C_2$, we select a centroid

$(c_{1i})_{i=1}^{m_1}$ and $(c_{2i})_{i=1}^{m_2}$, and compute the space folding measure $\chi$ between every pair of centroids, instead of using the original samples. By reducing $m_1$ and $m_2$, we can significantly lower the computational cost of calculating the measure $\chi$. The exact impact of this reduction is left for future work, however we provide preliminary results of impact of such grouping on the space folding values in Appendix C.

## 6 Experiments

### 6.1 Experimental Setup

**CantorNet.** We start the experimental evaluation of our measure on CantorNet, a hand-designed example inspired by the fractal construction of the Cantor set (Lewandowski et al., 2024) (see App. A for the summary). First, note that we may consider a subset of layers for the computations of the space folding measure;

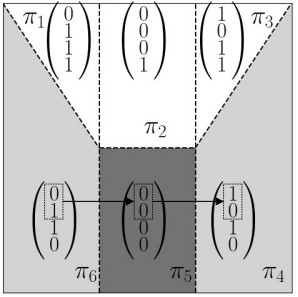 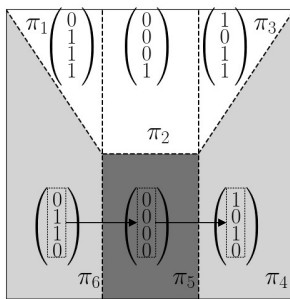

Figure 5: Activation patterns $\pi_i$ of the recursion-based representation of CantorNet. For the computation of the space folding measure $\chi$ we can consider a subset of layers; left: highlighted activations in the first layer, right: all layers. For a path $\Gamma : \pi_6 \to \pi_5 \to \pi_4$, the folding $\chi(\Gamma) = 0$ if we consider only the activations from the first layer, while $\chi(\Gamma) = \frac{1}{2}$ if we consider all the layers. Colours are used for increased visibility; "white" patterns form a convex set in the Hamming cube sense (see Ex. 2). We skip neurons with unchanged values.

however, we risk not detecting any folds. Indeed, for CantorNet of the recursion level $k = 1$, consider a path $\Gamma : \pi_6 \to \pi_5 \to \pi_4$ (see Fig. 5). If we consider only the activation patterns in the first layer, we obtain $\chi(\Gamma) = 0$, while if we include all the layers, then $\chi(\Gamma) = \frac{1}{2}$.

Next, we set the recursion depth to $k = 2$ (Fig. 6, the background), and create a path $\Gamma$ between points $\mathbf{x}_1 = (0, \frac{3}{4}), \mathbf{x}_2 = (1, \frac{3}{4})$ (Fig. 6, the arrows). We illustrate the evolution of range measures $r_1, r_2$ along path $\Gamma$ (Fig. 6, dotted and dashed curves, respectively), and the corresponding space folding measure (in blue). In this example, we obtain a higher space folding value than for the recursion level $k = 1$, $\chi(\Gamma) \approx 0.7$.

**MNIST.** Further, we study the behavior of the space folding measure on ReLU neural nets trained on MNIST (LeCun et al., 1998). We keep the number of hidden neurons constant (equal 60), and we experiment with the depth and the width of the network, trying the following architectures: $2 \times 30, 3 \times 20, 4 \times 15, 5 \times 12, 6 \times 10, 10 \times 6$, with the notation (no. layers)$\times$(no. neurons).[2] We then train those networks for 30 epochs on pre-defined random seeds (in Python NumPy and Torch), and store their parameters. For the networks that we managed to train to a high validation accuracy (especially for deeper networks it is highly dependent on the initialization), we analyze the relationship between the depth of a ReLU network and the aggregated median of maximas of non-zero space folding across all pairs of digits in the MNIST test set (100 pairs, $\sim 1M$ paths $\Gamma$ for each pair), obtaining the Pearson correlation coefficient 0.987 for the random seed (equal 4) where the networks performed the best. Interestingly, for the networks that did not reach satisfactory validation accuracy, the aggregated space folding was much lower than for networks that trained

---

[2]As underpinned by Lemma 1, we do not expect to see folding effects in a network with 1 hidden layer and 60 neurons and hence we omit it.

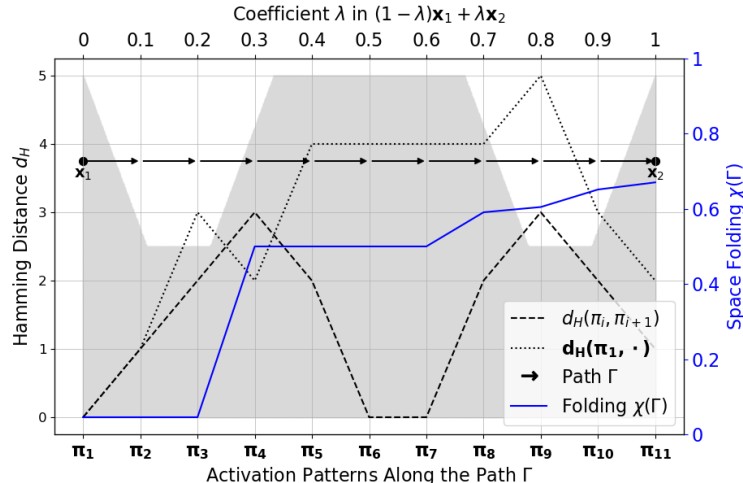

Figure 6: Behaviour of $d_H(\pi_1, \cdot)$ for respective $\pi_i$ (dotted line) and distance between neighboring patterns $d_H(\pi_i, \pi_{i+1})$ (dashed line) on path $\Gamma$ constructed between points $\mathbf{x}_1 = (0, \frac{3}{4})$ and $\mathbf{x}_2 = (1, \frac{3}{4})$ (indicated by arrows). Background represents CantorNet recursion-based representation at recursion level $k = 2$ (see App. A). For the illustration purposes, it has been scaled to fill the figure, but its domain is a unit square $[0,1] \times [0,1]$. The cumulative maximum (Eq. (2)) is $d_H(\pi_1, \pi_9) = 5$. Note that the Hamming distance $d_H$ between the initial activation pattern $\pi_1$ *can* decrease (dotted line), indicating deviations from convexity, as discussed previously. The blue line represents the space folding measure $\chi(\Gamma)$. (Best viewed in colors.)

well. It hints that the symmetries that arise with depth help with achieving some generalization ability. See Appendix B for more details.

We further perform similar experiments on much larger networks with $2 \times 300, 3 \times 200$, for no. layers $\times$ no. neurons. We observed that, while the folding values do not vary much, the ratio of paths that feature space folding increases dramatically: from $0.35 \pm 0.1$ and $0.44 \pm 0.15$ for networks $2 \times 30$ and $3 \times 20$, respectively, to $0.97 \pm 0.04$ and $0.99 \pm 0.02$ for networks $2 \times 300$ and $3 \times 200$, respectively. For more details see Appendix D.

## 6.2 Results

In Fig. 6, we have shown an interesting phenomenon. We first mapped a straight-line walk from the Euclidean input space to the Hamming activation space. During this mapping, while walking along the mapped path, we have sometimes observed a *decrease* in the Hamming distance with respect to the initial input point, while in the input space, the Euclidean distance is increasing. This indicates that there is a replication of the activation pattern along the path in the activation space, which we call *folding*. This is important, as it allows us to understand how neural networks transform and compress input data, revealing the intrinsic geometric properties of the network's activation space. In the next section, we provide more discussions and hypothesis for these results.

## 6.3 Discussion

**Beyond MLPs.** In the previous section, we have provided results on space folding by ReLU neural networks. So far, we have used all the hidden layers; however, a subset of hidden layers could also be utilized. We now provide additional remarks on the measure's utility when applied to different types of layers:

- **Residual layers**: Our study extends straightforwardly to networks with skip connections (e.g., ResNet (He et al., 2016)).
- **Normalization layers**: Normalization layers modify the coefficients of the weight matrices, which consequently alters the overall structure of the tessellation. However, our measure remains applicable as before. The exact impact of normalization is left for future study.

- **Attention layers**: While our measure may not be directly applicable to attention layers in its current form, it has been shown that ReLU can be used in self-attention mechanisms. Thus, our measure could potentially be applied to such layers (Shen et al., 2023).

## 7 Future Work

In this work, we have proposed a novel method to measure deviations from convexity between a walk on a straight line in the Euclidean input space and its mapping to the Hamming activation space. Further, we have used the introduced measure to analyze deviations from convexity under a synthetic example, CantorNet, and ReLU neural networks trained on MNIST. Our work highlights the convexity transformation of the input space by a ReLU neural network. In our experiments, we have tried various pairs of digits (intra- and inter-class) and observed qualitatively similar behavior across tested architectures: the max value of $\chi$ increases with the depth of the network, if it was trained to a high validation accuracy, and decreases if the achieved accuracy was lower. We thus hypothesize that the maximal value of $\chi$ is associated with the network's generalization capacity, as depth has been shown to be necessary (but insufficient) for generalization in neural networks by enabling deeper layers to learn hierarchical features (Telgarsky, 2015; 2016). We summarize our findings in the form of a hypothesis (The question) and its evidence (The answer).

> **The question:**
>
> Given a space folding value $\chi(\Gamma) = \tau \in [0, 1]$ for some path $\Gamma$, what can we learn?

> **The answer:**
>
> We propose to see the space folding measure $\chi$ as a *feature* of a neural network. It tells us how twisted the Euclidean input space becomes during the learning process. It is upper and lower bounded, thereby it may serve as a reference point for various models. As we have seen, on neural networks with a high degree of self-similarity (CantorNet), it can reach values close to 0.8, whereas for simple networks trained on MNIST to high validation accuracy the folding values are significantly lower. Moreover, the space folding measure $\chi$ provides insights into how effectively a network encodes self-similarities: higher values indicate more compact and efficient representations. Conversely, a low value suggests potential room for improvement in the network's architecture for the given problem.

We now outline several directions and discuss them briefly, with the hope that the community will build upon our efforts and advance these directions further. Although we consider this work to be a milestone that establishes a novel perspective on space transformation by neural networks, there are several important next steps we identify as future work.

Firstly, addressing the computational cost of our measure is an interesting and important direction. As we have already hinted in Sec. 5, one idea for future work is to explore how to optimize the choice of intermediate points on the path $\Gamma$ such that each intermediate point falls into a distinct linear region and the number of intermediate points $n$ equals the number of linear regions between the two samples. A greedy approach would be to use the upper bound $N$ on the number of linear regions between two input points $\mathbf{x}_1, \mathbf{x}_2$ (see the elaboration in Sec. 5) as follows: ($i$) interpolate $N$ points linearly (step 1 in Alg. 1), ($ii$) compare their activation patterns: if there are duplicates go back to ($i$) and sample more points, continue until all intermediate points have distinct activation patterns then break. A promising approach seems to be that proposed by Gamba et al. (2022), where the authors describe a method for discovery of linear regions between points $\mathbf{x}_1$ and $\mathbf{x}_1$ in the direction $\mathbf{d} = \mathbf{x}_2 - \mathbf{x}_1$. We remark that more efficient approaches may exist but we leave this exploration for future work. Furthermore, we will investigate reducing the computational cost of computing the measure $\chi$ by clustering samples within the classes $C_1$ and $C_2$. For preliminary experiments on the measure's sensitivity to that grouping see Appendix C.

Another possible next step, would be to investigate the folding effects between an image from a selected class and its adversarial perturbation across various attacks, as well as for different types of data augmentation techniques. It might also be worth examining empirically the space folding effects on various setups such

as networks trained with random labels, starting with no random labels, then progressing to 10% random labels, 20%, and up to 100%. Some natural follow-up questions that might arise of these experiments are to understand: What are the differences, What about different learning rates and optimization techniques, and How does the measure change when using a fraction of hidden neurons compared to all hidden neurons.

Another important and interesting direction to take on, would be to extend this study to various neural networks types $\mathcal{N}$ (e.g., binary neural networks (Courbariaux et al., 2015; Rastegari et al., 2016; Conti et al., 2018)), and transformer-like architectures with ReLUs (Shen et al., 2023; Mirzadeh et al., 2023).

Lastly, in the context of reinforcement learning, it has been shown that linear regions evolve differently depending on the policy used (Cohan et al., 2022). A natural extension of this work would be to investigate how our measure evolves under different policies.

## 8 Conclusions

We have proposed a novel method to measure deviations from convexity between a walk on a straight line in the Euclidean input space and its mapping to the Hamming activation space. Further, we have used the introduced measure to analyze deviations from convexity under a synthetic example, CantorNet, and ReLU neural networks trained on MNIST. Our work highlights the convexity transformation of the input space by a ReLU neural network.

## Acknowledgments

SCCH and JKU's research was carried out under the Austrian COMET program (project S3AI with FFG no. 872172), which is funded by the Austrian ministries BMK, BMDW, and the province of Upper Austria. We thank the anonymous reviewers for their constructive comments, which improved the quality of this work.

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

## A  CantorNet

CantorNet (Lewandowski et al., 2024) is a synthetic example inspired by the triadic construction of the Cantor set (Cantor, 1883). It features two representations opposite in terms of their Kolmogorov complexities, one linear in the recursion depth $k$, and one exponential. It is defined through the function $A : [0,1] \to [0,1] : x \mapsto \max\{-3x + 1, 0, 3x - 2\}$, as the *generating function* which is then nested as $A^{(k+1)}(x) := A(A^{(k)}(x))$, $A^{(1)}(x) := A(x)$. Based on the generating function, the decision manifold $R_k$ is defined as:

$$R_k := \{(x,y) \in [0,1]^2 : y \leq (A^{(k)}(x) + 1)/2\}. \tag{7}$$

The decision surface of $R_k$ (Eq. (7)) equals to the 0-preimage of a ReLU net $\mathcal{N}_A^{(k)} : [0,1]^2 \to \mathbb{R}$ with weights and biases defined as

$$W_1 = \begin{pmatrix} -3 & 0 \\ 3 & 0 \\ 0 & 1 \end{pmatrix}, b_1 = \begin{pmatrix} 1 \\ -2 \\ 0 \end{pmatrix}, W_2 = \begin{pmatrix} 1 & 1 & 0 \\ 0 & 0 & 1 \end{pmatrix} \tag{8}$$

and the final layer $W_L = \begin{pmatrix} -\frac{1}{2} & 1 \end{pmatrix}, b_L = \begin{pmatrix} -\frac{1}{2} \end{pmatrix}$. For recursion depth $k$, we define $\mathcal{N}_A^{(k)}$ as

$$\mathcal{N}_A^{(k)}(\mathbf{x}) := W_L \circ \sigma \circ g^{(k)}(\mathbf{x}) + b_L, \tag{9}$$

where $g^{(k+1)}(\mathbf{x}) := g^{(1)}(g^{(k)}(\mathbf{x})), \sigma$ is the ReLU function, and

$$g^{(1)}(\mathbf{x}) := \sigma \circ W_2 \circ \sigma \circ (W_1 \mathbf{x}^T + b_1). \tag{10}$$

## B    Heatmaps

In this section, we present additional results for the median of maxima of non-zero space folding values and the corresponding Median Absolute Deviation (MAD), defined as:

$$\text{MAD} := \text{median} \left( \sum_\Gamma |\chi(\Gamma) - \text{median}(\chi)| \right) \tag{11}$$

across different pairs of digits. We proceeded as follows:

---

**Algorithm 2:** Computation of Aggregated Folding Measures for Digit Pairs

---

**Input:** Images of digits from classes $C_1$ and $C_2$
**Output:** Median of non-zero maxima of space folding values across digit pairs $\pm$ MAD (Eq. (11))
**Step 1:** For each pair $(\mathbf{x}_{1i}, \mathbf{x}_{2j})$ where $\mathbf{x}_{1i} \in C_1$ and $\mathbf{x}_{2j} \in C_2$, compute $\chi(\Gamma)$ using Alg. 1.
**Step 2:** Among $|C_1| \cdot |C_2|$ measures, select those with non-zero values, forming the set $\chi(\Gamma)_+$.
**Step 3:** For each $\chi(\Gamma)_+$, compute the maximum value along $\Gamma$. Collect maximas into the set $\max \chi_+$.
**Step 4:** For each digit pair compute the median of $\max \chi_+ \pm$ MAD (Eq. (11)).

---

Each heatmap corresponds to a network with a different depth. We observe an increase in the median space folding values with the depth of the neural network. In the main paper, we reported the Pearson correlation coefficient between the depth and the aggregated median space folding $\chi$ for ReLU networks trained on the random seed (equal 4) for which the networks achieved high validation accuracy, even for deeper layers. Initially, we worked with seeds $\{0, 1, 2, 3, 4\}$, but only for seeds $\{3, 4\}$ did the network achieve the desired accuracy. For depths up to 5 layers, the validation accuracy easily exceeds 0.95, while for 6, and 10 layers, it drops, with the deepest network achieving around 0.85 of validation accuracy.

## C    Sensitivity to Grouping

In this section, we present a preliminary study on the sensitivity of our measure to clustered data as compared to working directly with the original samples. We study the effect of clustering ($i$) within one class of digits, ($ii$) within both classes of digits. Our analysis focuses on 16 pairs of digit classes from the MNIST test dataset (we work with digits from classes $\{0, 3, 6, 9\} \times \{0, 3, 6, 9\}$). We then vary the number of clusters ($k$) into which the digits are grouped, experimenting with $k \in \{1, 2, 5, 10, 20, 50, 100\}$. We use $k$-means in the (flattened) pixel space of the images. After clustering, the set of centroids is extracted and used as input to the procedure outlined in Algorithm 2. This process yields a single folding value for each digit pair. We repeat this analysis for all 16 digit combinations, resulting in 16 values for each level of clustering (x-axis), for which we report the mean and standard deviation values, and add the reference value of space folding for comparison purposes. We report the results for 2 different neural networks, distinguished by colors (Fig. 8).

Qualitatively, our results indicate that the behavior of the measure is not strongly affected even with as little as 5 clusters per group. Although these preliminary findings require validation on additional datasets, they suggest that computational complexity, as defined in Eq. (6), could be significantly reduced without compromising accuracy.

## D    Analysis for Deeper Networks

We performed a preliminary study on the folding effects for deeper networks with architectures $\{2 \times 300, 3 \times 200\}$ for the no. of layers $\times$ no. of neurons, respectively. As previously, we train those networks to

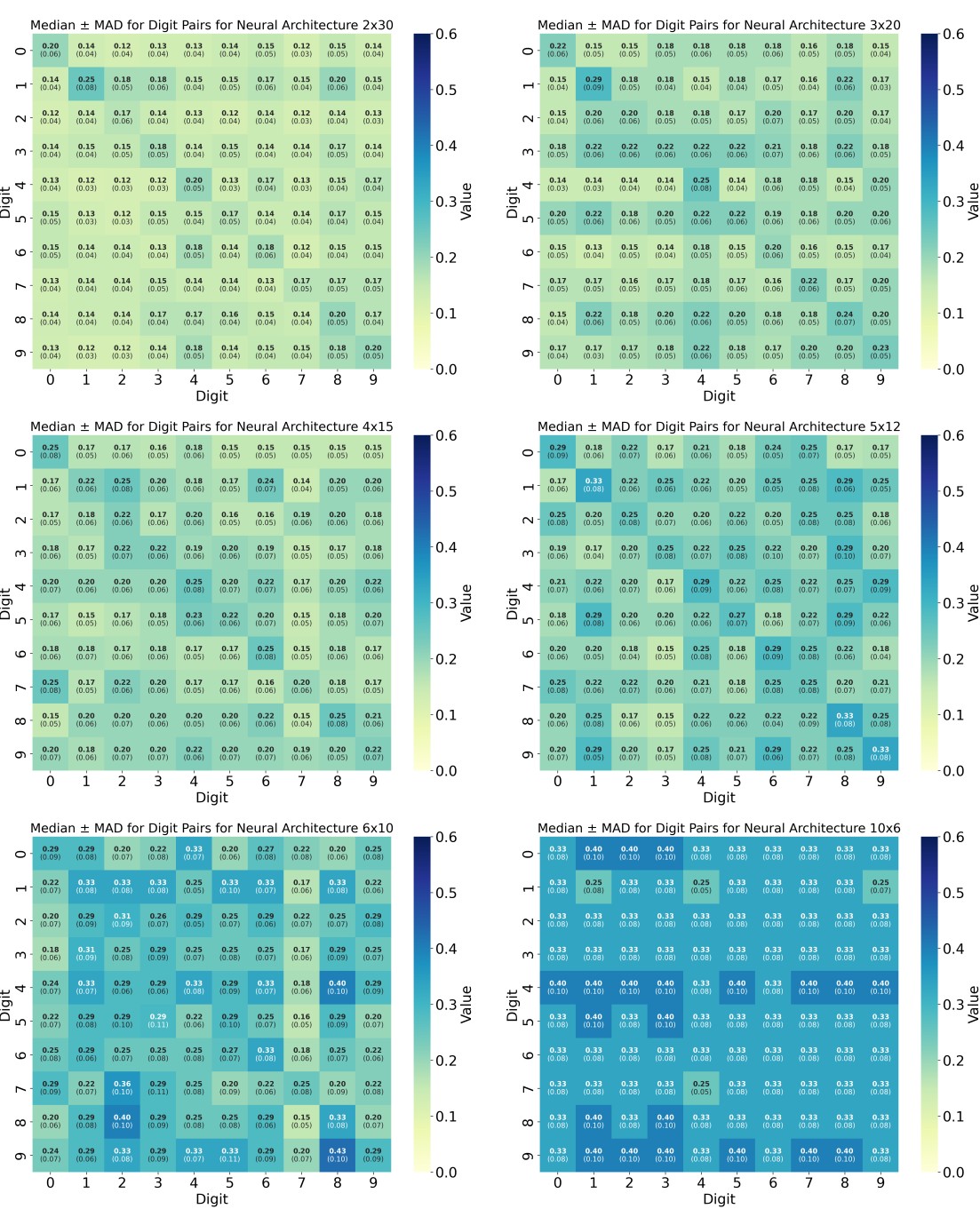

Figure 7: Median of maximas of non-zero space folding ± median absolute deviation for every digit pair in MNIST for a selected ReLU neural network trained with random seed 4. The space folding values increase with depth (darker blue values). (Best viewed in colors.)

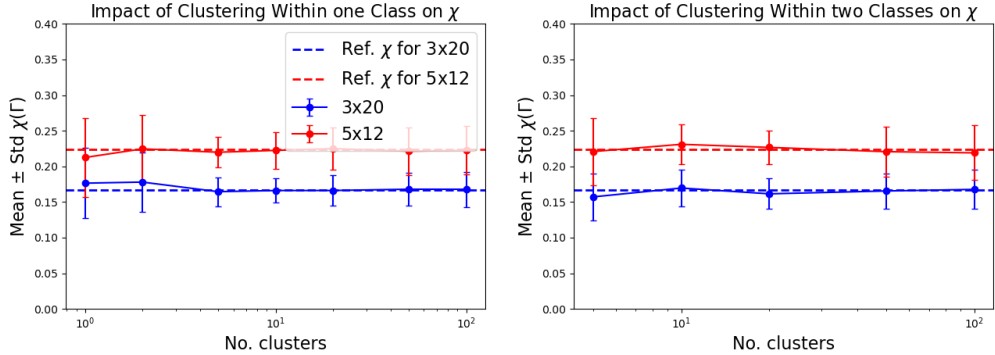

Figure 8: Preliminary results illustrate the robustness of the space folding measure $\chi$ under clustering. **Left:** Only one digit class $C_1$ is clustered, while the other class $C_2$ remains intact. The measure remains stable across varying numbers of clusters $k$. **Right:** Both digit classes are clustered. Although no folding effects arise for $k = 1, 2$, introducing $k = 5$ clusters per class already produces consistent folding effects, and these persist at higher clustering levels. Dashed lines indicate the reference values for the space folding measure computed using all the pairs of digits of respective classes.

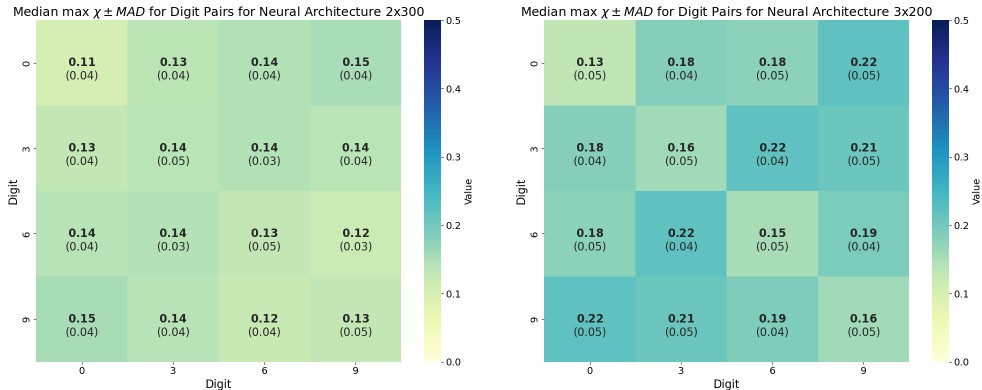

Figure 9: We investigate the folding effects for larger networks. Interestingly, we do not observe much difference for the folding values. However, we see a strong increase in the number of paths that feature folding effects.

convergence at MNIST, and store their parameters. We then investigate the space folding values similarly as before. Interestingly, we do not observe much difference in the space folding values as computed in Alg. 2 when compared to values obtained on shallower architectures (see Fig. 9). Although the folding values themselves show little difference compared to shallower networks, the proportion of paths exhibiting folding effects increases substantially. For the smaller architectures $2 \times 30$ and $3 \times 20$ the mean ratio ($\pm$ std) of paths with folding effects is $0.35 \pm 0.1$ and $0.44 \pm 0.15$, respectively, while for deeper architectures $2 \times 30$ and $3 \times 20$ those values are $0.97 \pm 0.04$ and $0.99 \pm 0.02$, respectively.

