# OpenReview forum: "On Space Folds of ReLU Neural Networks"
_TMLR — Accepted by TMLR_

### Review · Reviewer_sXC9 · 2024-11-24

**Summary Of Contributions:**

This paper studies the widely assumed property that ReLU neural networks rely on space folding transformations when learning. Compared to past works, this work is _the first to quantify the amount of space folding that occurs_. This quantitative measure is built by viewing the activation patterns as bits and quantifying the convexity ratio in the activation space when using Hamming distance as a distance function. Along a (shortest) path $\Gamma$ the space folding $\chi(\Gamma)$ is based on the ratio of two range metrics – the first measuring the maximum distance from the starting point; the second measuring the total distance travelled.

Experiments are obtained on synthetic models (CantorNet) and real world models (ones trained on MNIST). CantorNet is a network solving the decision problem of a point belonging to the Cantor set (right?) which is known to require high folding, and the experiment serves to validate our new metric. The MNIST experiment seems to affirm that deeper MLPs have higher folding.

**Audience:**

Yes

**Claims And Evidence:**

Yes

**Requested Changes:**

N.B. There are a few extra, regarding the clarity of the writing.

N.B.#2, I will use this space to get clarifications.

1. Example 2, Convexity. How did you get the intermediate states? E.g. for 0001->1011, a shortest path, flipping one bit at a time is: 0001->0011->1011 or 0001->1001->1011. Where did 1111 come from? Also, in this example why does the indexing start from 4 ($\pi_4$)? And shouldn’t these be added to the convex subset? (0011 is on a shortest path from two points in S, but is not in S, making S not be a convex subset)
1. What do the colours in Figure 2 represent? Please describe the figure. While I acknowledge that Lewandowski et al. have provided you with it, the paper is not publicly available yet. (Do send a link if it is).
1. [Nit] Convexity proof (=> direction). $R_{\pi_1}$ should be $R_P$? (similar for the other) Double-check and fix it if so.
1. [Nit?] Section 5, “The space folding measure”. $\Gamma$ suddenly gained a subscript $n$, but it doesn’t have one when introduced? Why? And is the subscript even necessary for the proof.
1. First experiment and Figure 5. Again, someone unfamiliar with Cantor set and CantorNet, like me, would find it very challenging to understand what is going on. What I understood (also from Wiki) is that:
   * The neural network has fixed weights and requires no training.
   * The “dataset” describes a fractal and the higher the recursion depth, the more complex (in terms of amount of folding) the neural network has to become.

   However, I do not know how was $\chi$ measured (did you sample random points on the input line?). Please, give a little more intuition about the task and let the reader know how the value was calculated. Also, the value of 0.5 is different from the value of 0.6+ in Figure 6, which confuses me more.
1. Figure 6 “Background represents CantorNet recursion-based representation at recursion level k = 2”. I find it quite hard to wrap my head around the sentence and the gray vs white background of the figure. Is the background related in any way to the x and y axis? Or, rather, just the x axis? (I presume the latter, but add some more clarification in the caption)
1. Figure 7. The standard deviations seem high (even looking at figure 8). To what quantitative extent, across the dataset, does the following correlation hold: “Deeper architectures have higher $\chi(\Gamma)$”. Given the use of word “significant”, I’d expect some correlation coefficients, at the very least. You can obtain those by:
    * considering different classes
    * retraining the networks with different random seeds

   I suggest you do both.
1. Address weaknesses.
   * state assumptions
   * add algorithm/complexity and a robustness figure (e.g. do your current observations change, if you linearly interpolate of steps 0.01 and 0.2)
   * (I will not get fixated on that one, but:) Consider adding one more dataset.
   * add future work discussion.

**Strengths And Weaknesses:**

## Strengths

1. (Good first impressions) While the paper operates with complex mathematical concepts, it was immediately obvious to me, a practitioner with a decent mathematical apparatus, that the authors have made a commendable effort in presenting the concepts and laying out the idea well. I appreciate it, as I would be struggling to review the paper if the majority of the concepts were just assumed to be known by the reader. I also appreciate the visualisations (Figs. 1, 3, 4). I do retain my reviewer's right to request some polishing/clarifications (cf. Requested changes), but I have seen papers that could present an equivalently hard topic with 0 images and worse flow.
1. (New metric) As the authors discuss in the related work section, the idea of viewing activations as bits/computation as space foldings/etc. is not new. However, I am also not aware of past work that quantifies the folding and if no other reviewer flags such a quantitative measure already existing, this would be a strong point for acceptance. It is not my subfield of research, but some googling got me nowhere close to a similar metric.

## Weaknesses
**The fact I listed only two strengths does not necessarily imply the paper does not have any others, or that it is bound to be rejected.**

(Allow me to put my practitioner’s hat on for most of these)

1. (Assumptions of the metric and generality) While I do not see why not, I also do not see the reason the paper focuses only on MLPs with ReLUs? After all, sections 4 and 5 mention ReLU only twice in total – once in figure 3, regarding the fact that hyperplanes intersect the entire input space (which the authors say it “holds for the hyperplanes [...] from the first hidden layer”) and once at the very end of \S4 (same statement). Hence, I have to ask the question, would the metric be of any use, if we have:
     * normalisation layers – I presume not, as they may break the straightness of the lines? (unsure)
     * residual connections – I cannot see why not
     * attention layers – except for the attention coefficients' calculations, the rest of the dataflow are affine transformations.

   I believe the assumptions behind this metric (and reasons to focus on MLPs) should be emphasised in \S3 and 4. While I know it is common for theory and especially interpretability to lag behind practical advances, if this study is not easily extensible to other architectures, I would consider it as a ground for rejection.

1. (Measure algorithm, computational complexity and robustness) The authors currently state that “computing the folding measure for large architectures may seem infeasible due to the potentially high number of linear regions”. However, the authors never stated the exact algorithm to calculate the measure or its time complexity. From my understanding, the steps, **for two points** are:
   * Interpolate them linealry. Sample $n$ points on this interpolation. $O(n)$
   * For each point, compute the binarisation. $O(n*Cost)$ where $Cost$ is the complexity to run the network in inference mode.
   * Calculate the maximal distance from the starting point and the sum of distances between intermediate points. As we are using Hamming distance, for $N$ neurons this is $O(N)$ for one measure. We need $O(n)$ such measures, so $O(n*N)$ in total.

   If my algorithm is correct, this is $O(n*(N+Cost))$ for a *pair of samples*. Measuring the distance, as done in the experiments, for one **pair of classes** $C\_1$ and $C\_2$ is $O(n*(N+Cost)*|C_1|\*|C_2|)$.

   Obviously, this will not easily scale. As we cannot change the $(N+Cost)$ factor, we need subsampling. This can be achieved by varying $n$ or subsampling from the classes. I, however, have zero idea and, unfortunately, zero evidence of how robust the proposed measure is to varying those hyperparameters.
1. (Few experiments) Currently, the measure is only evaluated on two datasets, and for MNIST, only on one pair of classes. Consider, e.g. making a heatmap for all pairs of classes? E.g. a board with 5 10x10 (or even 5x5) heatmaps for each architecture, all sharing the same legend (so as to show that $\chi$ is indeed greater for all pairs for the 6x10 architecture). Feel free to consider other datasets too.
1. (What can we learn) The authors say that $\chi$ should be seen “as a feature of a neural network”. I have two problems with this, one practical and one philosophical:
   * The measure is also based on the data. Thus, it is rather a feature of the network and the data as a pair.
   * What leverage do I get, from calculating it? Do I get to understand what the “underlying logic” of the model is? Or can it tell me how to improve the architecture? Yes, I know it measures the amount of twisting/folding the NN will do to an input, but adding some discussion of future directions, apart from the fact you will investigate the space folding measure of more pretrained models, would likely improve your paper on _”Would some individuals in TMLR's audience be interested in the findings of this paper?”_ aspect of the paper.

---

> ### Author Response · Authors · 2024-12-05
> **In this answer, we address the requested changes part of the review.**
>
> We thank the reviewer for the valuable time and feedback. We have included all the below changes to the manuscript and marked them in a blue color.
>
> 1. The reviewer is right about the activation patterns. There was an error on our side: (1111) is on the path in the case (3), but not (2), and in the case (2) there is (0011) which was previously found in (3). We have corrected the example in the paper. Moreover, we have updated the labeling of activation patterns: what has been  {$\pi_4,\pi_5,\pi_6$} is now {$\pi_1, \pi_2, \pi_3$}.
>
> With respect to breaking the convexity of the “white set” by (0011): if we consider the set of non-observable patterns (we refer to the activation regions with newly labeled activation patterns)  $S:=${$\pi_1, \pi_2, \pi_3$}. Observe that none of the activation patterns $\pi_{non-obs}$, which is on the shortest path between activation pattern in $S$, is included in {$\pi_4,\pi_5,\pi_6$} and thus the set $S$ is convex in the sense of Def. 1.
>
>
> 2. We thank the reviewer for the suggestion. We have extended the figure's caption.
>
>
> 3. The reviewer is right: in the proof, in the direction “=>” $R_{P_1}$ is the same as $R_P$. However, as we have more activation patterns in that figure, we opted for a consistent notation. We do acknowledge that it might introduce some redundancy in the notation, and if the reviewer believes that changing $R_{P_1}$ to $R_P$ improves readability, we will do it.
>
>
> 4. $\Gamma$ gained a subscript $n$: Corrected. The intention was to denote the number of steps on the path \Gamma, but the reviewer is right that it is better to remove the subscript.
>
>
> 5. We have contacted Lewandowski et al with the request to submit the article to Arxiv. They shared with us the following link: https://arxiv.org/abs/2411.19713
>
> We measured $\chi$ between $\mathbf{x}_1 = (0,¾)$ and $\mathbf{x}_2=(1,¾)$.  We have included it in the figures caption, thanks for the suggestion.
>
> The difference in the values between the Fig.6: Note that the recursion level $k$ of CantorNet is different: in Fig. 5 we set $k=1$ and in Fig.6 we have $k=2$.
>
>
> 6. We thank the reviewer for pointing it out. By “Background represents CantorNet recursion-based representation at recursion level k = 2” we meant that from two alternative yet equivalent representations of CantorNet we chose the one Lewandowski et al. refer to as “recursion-based” (please see the CantorNet paper, Sec.3.1 or Appendix A in our paper). Then, we set $k=2$, which Lewandowski et al. refer to as a “recursion level”. We have improved the caption.
>
>
> 7. We have run additional experiments and added the results to the paper. As the reviewer suggested, we considered different classes in MNIST and retrained the networks with different random seeds. See this point addressed in Weaknesses Section (and Appendix B in the updated paper).

---

> ### Author Response · Authors · 2024-12-05
> **In this answer, we address the weaknesses part of the review p1**
>
> 1. (Assumptions of the metric and generality) We have added the following paragraph to the Sec.6
>
> “Beyond MLPs. In the previous section, we have provided results on space folding by ReLU neural networks. So far, we have used all the hidden layers; however, a subset of hidden layers could also be utilized. We now provide additional remarks on the measure's utility when applied to different types of layers:
>
> - Residual layers: Our study extends straightforwardly to networks with skip connections (e.g., ResNet He et al.).
> - Normalization layers: Normalization layers modify the coefficients of the weight matrices, which consequently alters the overall structure of the tessellation. However, our measure remains applicable as before. The exact impact of normalization is left for future study.
> - Attention layers: While our measure may not be directly applicable to attention layers in its current form, it has been shown that ReLU can be used in self-attention mechanisms Shen et al. (2023). Thus, our measure could potentially be applied to such layers.”
>
> Additional references in the order of appearance:
>
> - He et al., 2016.  “Deep residual learning for image recognition”. In Proceedings of the IEEE conference on computer vision and pattern recognition,
> - Shen et al., 2023. “A study on relu and softmax in transformer.” In: Arxiv.
>
>
>
> 2. (Measure algorithm, computational complexity and robustness)  We thank the reviewer for the valuable input! We included the mentioned algorithm (slightly modified) in Section.5, after introducing the measure itself. After the algorithm itself, we have added the following:
>
> “The complexity of running the network in inference mode is denoted as $O(\texttt{C})$. The total computational complexity for every pair of samples from classes $C_1$ and $C_2$ is thus
> $$
> O\left(n \cdot (N+\texttt{C})\cdot |C_1| \cdot |C_2|\right),
> $$
> where $|C_1|,|C_2|$ denote the cardinalities of the respective classes of points $\mathbf{x}_1,\mathbf{x}_2$.
> Since $N+\texttt{C}=\text{const}$, the overall computational cost can be controlled by adjusting $n$ or by subsampling within the classes. In the following, we establish an upper bound on the number of intermediate steps $n$. For a neural network with $N$ hidden neurons, consider two input samples $\mathbf{x}_1$ and $\mathbf{x}_2$. The maximum Hamming distance between their corresponding activation patterns, $\pi_1$ and $\pi_2$, equals $d_H(\pi_1,\pi_2)=N$. Consequently, the number of linear regions between inputs $\mathbf{x}_1$ and $\mathbf{x}_2$ is at most $N$. Since intermediate steps that fall within the same activation region do not affect the space folding measure, the number of steps $n$ should not exceed $N$.
> We  note that evenly spacing intermediate points for the computation of the space folding measure $\chi$ may be suboptimal. Instead, the focus should be on traversing \emph{all} linear regions between the two samples. Optimization of the path will be addressed in a future submission.
>
> Secondly, to address the (potentially) high cardinality of classes $C_1$ and $C_2$, we propose the following approach. Let $m_1 < |C_1|$ and $m_2 < |C_2|$, where $m_1$ and $m_2$ represent the number of clusters into which the samples in classes $C_1$ and $C_2$ are grouped. For each cluster in respective class $C_1$ or $C_2$, we select  centroids $(c_{1i})$ and $(c_{2i})$, and  compute the space folding measure $\chi$ between every pair of centroids, instead of using the original samples. By reducing $m_1$ and $m_2$, we can significantly lower the computational cost of calculating the measure $\chi$. The exact impact of this reduction is left for future work.”
>
>
> 3. (Few experiments) We have run additional experiments and added the results to the paper. As the reviewer suggested, we considered all digit pairs in MNIST and retrained the networks with different random seeds. The heatmaps of aggregated values of space folding are in Appendix B. We have further removed the figure showing the evolution of the space folding measure between digits 6 and 9, and instead reported the correlation coefficient between the network’s depth and the aggregated median of space folding for every digit pair for networks trained with 2 different random seeds. In the Appendix B, we append detailed results.

---

> ### Author Response · Authors · 2024-12-05
> **In this answer, we address the weaknesses part of the review p2**
>
> (What can we learn) We have refined the “The Answer” colorbox part in the Discussion section (S.7), adding the following:
>
> “Moreover, the space folding measure $\chi$ provides insights into how effectively a network encodes self-similarities: higher values indicate more compact and efficient representations. Conversely, a low value suggests potential room for improvement in the network's architecture for the given problem.”
>
> We have also added the section Future Work with the following content (now Section 7):
>
>
> “Firstly, we will address the computational cost of our measure. As hinted in Sec. 5, we will explore how to optimize the choice of intermediate points on the path $\Gamma$ such that each intermediate point falls into a distinct linear region and the number of intermediate points n equals the number of linear regions between the two samples. A greedy approach would be to use the upper bound N on the number of linear regions between two input points $x_1, x_2$ (see the elaboration in (Measure algorithm, computational complexity and robustness) ) as follows: (1) interpolate N points, (2) compare their activation patterns: if there are duplicates go back to (1) and sample more points, continue until all intermediate points have distinct activation patterns then break. We remark that perhaps more efficient methods exist but we leave this exploration for future work.
> Additionally, we will investigate reducing the computational cost of computing the measure $\chi$ by clustering samples within the classes $C1$ and $C2$.
>
>
> Secondly, we will investigate the folding effects between an image from a selected class and its adversarial perturbation across various attacks, as well as for different types of data augmentation techniques. It might also be worth examining the space folding effects on networks trained with random labels, starting with no random labels, then progressing to 10% random labels, 20%, and up to 100%. What are the differences? What about different learning rates and optimization techniques? How does the measure change when using a fraction of hidden neurons compared to all hidden neurons?
>
> Thirdly, we will extend our study to various neural networks N (e.g., binary neural networks (Courbariaux et al., 2015; Rastegari et al., 2016; Conti et al., 2018)), and transformer-like architectures with ReLU activation functions (Shen et al., 2023; Mirzadeh et al., 2023) and record their global space folding measure, $\Phi$, as defined in Eq. (5).
>
>  Lastly, in the context of reinforcement learning, it has been shown that linear regions evolve differently depending on the policy used (Cohan et al., 2022). A natural extension of our work would be to investigate how our measure evolves under different policies”
>
> Additional references in the order of appearance:
>
> - Courbariaux et al., 2015. “Binaryconnect: Training deep neural networks with binary weights during propagations.” In: NeurIPS.
> - Rastegari et al., 2016. “Xnor-net: Imagenet classification using binary convolutional neural networks.” In: ECCV.
> - Conti et al., 2018. “Xnor neural engine: a hardware accelerator ip for 21.6 fj/op binary neural network inference.” In: IEEE Transactions on Computer-Aided Design of Integrated Circuits and Systems.
> - Shen et al., 2023. “A study on relu and softmax in transformer.” In: Arxiv.
> - Mirzadeh et al., 2023. “Relu strikes back: Exploiting activation sparsity in large language models.”  In: Arxiv.
> - Cohan et al., 2022. “Understanding the evolution of linear regions in deep reinforcement learning.” In: NeurIPS.

---

> ### Comment · Reviewer_sXC9 · 2024-12-05
> **Much better**
>
> Dear authors,
>
> Having read the updated parts of the paper and your rebuttal I am fairly satisfied with the current state of the paper. Particularly, the claims are more accurate and the future work discussion gives me enough hope there will be audience for this paper within TMLR.
>
> **I am therefore willing to provisionally increase my scores and recommend acceptance**
>
> That being said, I believe you have left an experiment that is seemingly "simple" (to my understanding) "for future work". Oh, I also need one clarification, that you might have omitted from your reply.
>
> Starting with the clarification, as it is shorter: In Figure 6, the line for Folding (the blue one) -- is this the folding between $\pi_1$ and $\pi_j$ for $j \in \{2\dots11\}$? If so, isn't the folding for $x_1$ and $x_2$ equal to a value between 0.6 and 0.8, not 0.5?
>
> Regarding future work, I understand achieving the most optimal sampling strategy is non-trivial and I'm happy to leave it out. However the clustering strategy you suggest seems both interesting and easy to compute. You can still leave **exact** impact for future work, but I believe having a preliminary proof-of-concept experiment, with just a couple of pairs of classes, would strengthen the contributions of the paper.

---

> > ### Author Response · Authors · 2024-12-08
> > **clarification & the impact of the clustering on folding**
> >
> > Question raised by the reviewer:
> > In Figure 6, the line for Folding (the blue one) -- is this the folding between $\pi_1$ and $\pi_j$ for $j=2,...,11$? If so, isn't the folding for $x_1$ and $x_2$  equal to a value between 0.6 and 0.8, not 0.5?
> >
> > Our answer (clarification):
> > We thank the reviewer for pointing this out. Please find all updates in purple in our manuscript. Yes, the reviewer is right: the folding value is between 0.6 and 0.8. We believe that by mentioning  0.5 for the value of folding, the reviewer is referring to the remark that considers taking a subset of layers for the computation of the measure. The two values are reported for different recursion-levels of CantorNet - for $k=1$ we obtain $\chi(\Gamma=1/2$, while it is higher for $k=2$. We have improved the explanation in that paragraph, and hope that it is more clear now.
> >
> >
> > Question raised by the reviewer:
> >
> > However the clustering strategy you suggest seems both interesting and easy to compute. You can still leave exact impact for future work, but I believe having a preliminary proof-of-concept experiment, with just a couple of pairs of classes, would strengthen the contributions of the paper.
> >
> >
> > Our answer:
> >
> > We thank the reviewer for encouraging us to still include it in our paper! Please find all updates in purple in our manuscript. We have run a preliminary analysis investigating the sensitivity of our measure to clustering within classes, and using cluster centroids to compute the measure instead of the original data samples. We have included the results in Appendix C: qualitatively, we found that the behaviour of our measure remains largely consistent across various numbers of clusters; for more details please see Appendix C.
> >
> > Additionally, we have run some experiments on larger networks as suggested by another reviewer, and included them in Appendix D. We have also included additional clarifications in other sections, all in the purple color. The reviewer might be interested to have a look.

---

> > > ### Comment · Reviewer_sXC9 · 2024-12-08
> > > **Potential improvement for the clustering figure**
> > >
> > > Dear authors,
> > >
> > > The figure looks good. I was just wondering if it would help to display the actual folding value (e.g. as flat dashed line) when no clustering is used?

---

> > > > ### Author Response · Authors · 2024-12-09
> > > > **included**
> > > >
> > > > We thank the reviewer for the suggestion. We have included the actual folding value as a dashed line in consistent colors for each architecture considered (we have removed the median values displayed previously).  We have also updated the paper’s contributions, and the organization of the paper.  Please find all updates in red in our manuscript.

---

> > > > > ### Comment · Reviewer_sXC9 · 2024-12-09
> > > > > **LGTM**
> > > > >
> > > > > LGTM

---

> > > > > > ### Author Response · Authors · 2024-12-20
> > > > > >
> > > > > > We thank the reviewer for help in improving our paper.

---

### Review · Reviewer_jq1r · 2024-12-05

**Summary Of Contributions:**

This paper considers the problem of how ReLU neural networks fold input spaces, which is beneficial for users to understand the working mechanism of the networks. The paper proposes a novel measure building on the range measure which can measure the fold-ness of the resulting space and demonstrate a connection between the convexity of the input space and the activation space. Empirical evidence was provided on both CantorNet and MNIST image classification benchmark.

**Audience:**

Yes

**Broader Impact Concerns:**

The work is largely theoretical, thus not applicable.

**Claims And Evidence:**

Yes

**Requested Changes:**

The authors reported the computational complexity of their algorithm, which, to me, is not that high when C_{1} and C_{2} are relatively small. In the experiment section, for MNIST classification the employed fully connected neural network has fewer parameters than one might typically expect. While this is not critical, it would be interesting to observe what happens with a larger number of neurons, e.g. three layers with 200 units per hidden layer; or would this be too computationally challenging?

**Strengths And Weaknesses:**

I am not an expert for this task. P.S. before I sent my review I noticed that the authors added a revision which already addressed some of the issues that I was observing.

Strengths:

The question is interesting from a theoretical point of view. The paper is generally well-written, and contexts are provided to help the readers understand the topic.

Weaknesses:

I find definition 1 and convexity and example 2 confusing. Definition 1 states that a subset is convex if all points on every shortest path between every two points are also in the subset. In example 2, it was stated that there are intermediate patterns, which I think lies on shortest path(s), not in the subset; yet the subset still forms a convex set, and the author justified that by stating that no intermediate points are contained in the complement of the subset. Does this contradict with the definition? Or am I understanding it incorrectly?

The proposed measure needs to be computed between every pair of input samples, thus leading to relatively high computational cost. Is it in principle possible to calculate the measure based on a batch of pairs?

---

> ### Author Response · Authors · 2024-12-08
> **We thank the reviewer for their valueable time and feedback.**
>
> Question raised by the reviewer:
>
> I find definition 1 and convexity and example 2 confusing. Definition 1 states that a subset is convex if all points on every shortest path between every two points are also in the subset. In example 2, it was stated that there are intermediate patterns, which I think lies on shortest path(s), not in the subset; yet the subset still forms a convex set, and the author justified that by stating that no intermediate points are contained in the complement of the subset. Does this contradict with the definition? Or am I understanding it incorrectly?
>
>
>  Our answer:
>  We thank the reviewer for pointing it out. Please find all updates in purple in our manuscript. We have added a clarification to the definition that one should consider “observable” activation regions, and a footnote that gives an example for Fig. 1, left, where the region (011) is non-observable. We have also improved the explanation in example 2, which now reads as follows:
>
> “Consider activation patterns $\pi_1=(0111), \pi_2=(0001), \pi_3=(1011)$ as shown in Fig.2. For a walk through any two of the activation patterns,  (1) $\pi_1\to\pi_2$, (2) $\pi_2\to\pi_3$, (3) $\pi_1\to\pi_3$, there are intermediate  activation patterns $\pi_{inter}$ that we traverse. They are, respectively:  (1) $\pi_{inter}=\{(0011), (0101)\}$, (2) $\pi_{inter}=\{(0011), (1001)\}$, (3) $\pi_{inter}=\{(0011), (1111)\}$. As none of them is contained in $\{\pi_4,\pi_5,\pi_6\}$, we conclude that they are non-observable on the considered domain, $[0,1]\times[0,1]$. Thus, the activation patterns $\{\pi_1, \pi_2, \pi_3\}$ form a convex set in the Hamming cube sense. “
>
> Question raised by the reviewer:
>
> The proposed measure needs to be computed between every pair of input samples, thus leading to relatively high computational cost. Is it in principle possible to calculate the measure based on a batch of pairs?
>
>
>
> Our answer:
>
> We thank the reviewer for the question. Please find all updates in purple in our manuscript. We have run a preliminary analysis investigating the sensitivity of our measure to clustering within classes, which is computable in conditions such as per-batch, and using cluster centroids to compute the measure instead of the original data samples. We have included the results in Appendix C.
>
> Requested changes from the reviewer:
>
>
> The authors reported the computational complexity of their algorithm, which, to me, is not that high when C_{1} and C_{2} are relatively small. In the experiment section, for MNIST classification the employed fully connected neural network has fewer parameters than one might typically expect. While this is not critical, it would be interesting to observe what happens with a larger number of neurons, e.g. three layers with 200 units per hidden layer; or would this be too computationally challenging?
>
>
>
> Our answer:
> We thank the reviewer for the suggestion. Please find all updates in purple in our manuscript. We have run the requested experiments for two bigger networks, 2 layers with 300 neurons each and 3 layers with 200 neurons each. We have included the results in Appendix D. Interestingly, we found that the behaviour of the measure is not largely affected; however, the ratio of paths that feature the folding effects dramatically increases. For more details please see Appendix D.

---

> > ### Author Response · Authors · 2024-12-20
> >
> > We are looking forward to the reviewer's feedback and we wish the reviewer happy holidays!

---

> > > ### Comment · Reviewer_jq1r · 2024-12-20
> > >
> > > I thank the authors for engaging in discussions. I think the paper is now in a good shape.

---

> > > > ### Author Response · Authors · 2024-12-20
> > > >
> > > > We thank the reviewer for help in improving our paper.

---

### Review · Reviewer_ho1r · 2024-12-08

**Summary Of Contributions:**

This paper introduces a concept called the "space folding measure" to analyze how ReLU networks transfer convexity between the Euclidean input space and the Hamming activation space. The authors show that straight-line paths in the input space correspond to non-convex trajectories in the activation space and propose range-based metrics to quantify these deviations. Their empirical analysis includes experiments with a synthetic "CantorNet" architecture as well as networks trained on the MNIST dataset to validate their findings.

**Audience:**

Yes

**Claims And Evidence:**

Yes

**Requested Changes:**

Please see 'weaknesses'.

**Strengths And Weaknesses:**

Strengths:

1. The paper investigates an interesting property of ReLU networks and supports its findings with empirical analysis.

2. Practical implications of the proposed measure are discussed, particularly in Section 7.


Weaknesses:

The paper's organization could be improved. While many parts of the paper have been updated, the introduction and abstract do not reflect these revisions. These sections should be updated to align with the rest of the paper and provide a clearer overview of the contributions.

---

> ### Author Response · Authors · 2024-12-09
> **Updated**
>
> We thank the reviewer for pointing it out. Please find all updates in red in our manuscript. We have updated the paper’s contributions, and the organization of the paper.

---

> > ### Author Response · Authors · 2024-12-20
> >
> > We are looking forward to the reviewer's feedback and we wish the reviewer happy holidays!

---

> ### Author Response · Authors · 2025-01-10
>
> This is a friendly reminder that we are still awaiting the reviewer's recommendation on our paper. As we are currently preparing another submission, having the decision would be greatly beneficial, as we would like to reference the outcomes of this paper. We sincerely thank the reviewer for the time and effort in reviewing our work.

---

### Author Response · Authors · 2025-02-10
**Comment on the Camera-Ready Version**

We would like to express our gratitude to the reviewers for their valuable feedback, which has helped us improve our paper. In our camera-ready, we have improved the presentation of the additional material added during the rebuttal to make the paper more coherent, and have recreated the heatmaps in appendix C using a better performing mode.

---

### Decision · Action_Editor_Fp7G · 2025-01-08

**Recommendation:** Accept as is

**Comment:**

The Authors have addressed all of the major issues highlighted by the Reviewers, and all Reviewers are in favour of accepting the work to TMLR. Well done!

**Audience:**

This is a well-presented idea likely to be useful to a significant proportion of TMLR's readership, given that it focuses on a class of neural networks with ReLU activations that remain highly popular in the community.

**Claims And Evidence:**

Quantifying space folding induced by ReLU neural networks is a valuable and novel direction, which the authors present and defend convincingly enough to meet the bar for TMLR.